# A Preliminary Investigation on the Functional Validation and Interactions of *PoWOX* Genes in Peony (*Paeonia ostii*)

Mengsi Xia [1,2,†], Wenbo Zhang [1,2,†], Yanting Chang [1,2], Yanjun Ma [1,2], Yayun Deng [1,2], Keke Fan [1,2], Xue Zhang [1,2], Zehui Jiang [1,2,*] and Tao Hu [1,2,*]

1    International Center for Bamboo and Rattan, No. 8, Futong Eastern Avenue, Wangjing Area, Chaoyang District, Beijing 100102, China; xiamengsi@icbr.ac.cn (M.X.); wenbozhang@icbr.ac.cn (W.Z.); changyanting@icbr.ac.cn (Y.C.); mayanjun@icbr.ac.cn (Y.M.); yayundeng@icbr.ac.cn (Y.D.); fankk@icbr.ac.cn (K.F.); zx949949@126.com (X.Z.)
2    Key Laboratory of National Forestry and Grassland Administration/Beijing for Bamboo & Rattan Science and Technology, No. 8, Futong Eastern Avenue, Wangjing Area, Chaoyang District, Beijing 100102, China
*    Correspondence: jiangzh@icbr.ac.cn (Z.J.); hutao@icbr.ac.cn (T.H.)
†    These authors contributed equally to this work.

**Abstract:** As a woody plant, peony (*Paeonia suffruticosa*) has a long growth cycle and inefficient traditional breeding techniques. There is an urgent need in peony molecular breeding to establish an efficient and stable in vitro regeneration and genetic transformation system, in order to overcome the recalcitrant characteristics of peony regeneration and shorten the breeding cycle. The development of plant somatic embryos is an important way to establish an efficient and stable in vitro regeneration and genetic transformation system. Plant-specific WUSCHEL-related homeobox (WOX) family transcription factors play important roles in plant development, from embryogenesis to lateral organ development. Therefore, in this research, four *PoWOX* genes of "Fengdan" (*Paeonia ostii*) were cloned from the peony genome and transcriptome data of preliminary peony somatic embryos. The sequence characteristics and evolutionary relationships of the *PoWOX* genes were analyzed. It was demonstrated that the four *PoWOX* genes, named *PoWOX1*, *PoWOX4*, *PoWOX11*, and *PoWOX13*, belonged to three branches of the WOX gene family. Their expression patterns were analyzed at different stages of development and in different tissues of peony seedlings. The expression localization of the *PoWOX* genes was determined to be the nucleus via subcellular localization assay. Finally, the interaction protein of the *PoWOX* genes was identified via yeast two-hybrid assay combined with bimolecular fluorescence complementation assay. It was shown that PoWOX1 and PoWOX13 proteins could form homodimers by themselves, and PoWOX11 interacted with PoWOX1 and PoWOX13 to form heterodimers. Peony stem cell activity may be regulated from PoWOX1 and PoWOX13 by forming dimers and moving to peony stem cells through plasmodesmata. Additionally, PoWOX11–PoWOX1 and PoWOX11–PoWOX13 may play important regulatory functions in promoting the proliferation of stem cells and maintaining the homeostasis of stem cells in the SAM of peony stems. Exploring the critical genes and regulatory factors in the development of the peony somatic embryo is beneficial not only to understand the molecular and regulatory mechanisms of peony somatic embryo development but also to achieve directed breeding and improvements in efficiency through genetic engineering breeding technology to accelerate the fundamental process of molecular breeding in peony.

**Keywords:** Peony (*Paeonia ostii*); *PoWOX*; subcellular localization; yeast two-hybrid; gene interaction

## 1. Introduction

Peony (*Paeonia suffruticosa*) is native to China and belongs to the peony group (Section Moutan DC) of the genus *Paeonia* in the family Paeoniaceae. It is one of the ten most famous traditional flowers in China, famous for its large and colorful flowers, as well as an important economic plant with both medicinal and oil value [1–5]. The long growth

cycle and low efficiency of traditional breeding techniques of peony have severely limited the research on new variety breeding and molecular breeding of peony [6,7], resulting in a contradiction between the demand for new peony varieties with "novel" characteristics in the peony market and the insufficient renewal of existing peony varieties in terms of important ornamental traits [8,9]. Therefore, there is an urgent need in peony molecular breeding to establish an efficient and stable in vitro regeneration and genetic transformation system in order to overcome the recalcitrant characteristics of peony regeneration and shorten the breeding cycle.

The WUSCHEL-related homeobox (WOX) family is a superfamily of homeobox (HB) transcription factors in eukaryotes. It belongs to a plant-specific transcription factor, and all members of this family contain a conserved homeodomain (HD). The HD can be recognized and bound by specific DNA sequences and consists of 60–66 residues folded into a helix-turn-helix structure [10–12]. After a comprehensive study of the *Arabidopsis* genome, Haecker found that the *Arabidopsis* WOX family contains a total of 15 genes [13]. These genes were classified into three clades: the modern/WUS clade (WUS and WOX1-7), the intermediate clade (WOX8-9 and WOX11-12), and the ancient clade (WOX10 and WOX13-14). In addition to HD, members of each clade also contain other functional elements. For example, members of the modern/WUS clade contain a specific WUS-box (T-L-X-L-F-P-X-X, X represents any amino acid) with TL as the initial amino acid, but the starting amino acid is not fixed in the ancient clade and intermediate clade [12].

Studies have shown that the members of the WOX family have a wide range of functions and play an important role in the maintenance of the apical meristem tissue and stem cells, the formation of lateral and floral organs, embryo development, hormone signaling, and resistance metabolism in plants [13–19]. In particular, they play an important role in regulating the proliferation and differentiation of cells [19]. The *WOX1* gene regulates leaf development, the *Arabidopsis AtWOX1* gene mainly regulates the proliferation of lateral organs, and *AtWOX1* and *AtWOX3* together regulate the lateral growth of leaves [20,21]. The *WOX1* homologous gene *SlLAM1* in tomatoes promotes several types of leaf expansion and regulates leaf outgrowth, especially mid-lateral axis leaves, as well as the initiation of secondary leaflet growth; it also acts on the growth of floral organs and affects the fertility of gametophytes [22,23]. *Arabidopsis AtWOX4* promotes the development of the primordial formative layer [24], regulates the lateral growth of plants [25], and is involved in regulating vascular cell division [26]. Poplar *WOX11/12a* promotes salt tolerance in poplar by enhancing ROS scavenging [27], and *WOX11* in rice recruits histone H3K27me3 demethylase to promote gene expression related to rice shoot development [28]. The *AtWOX13* gene is involved in membrane formation during fruit development [29], and wound-induced *AtWOX13* in *Arabidopsis* plays a role in callus formation and organ reconnection [30]. *AtWOX14* is mainly involved in the lignification process of *Arabidopsis* [31].

In ornamental plants, overexpression of the *Rosa canina RaWUS* gene was found to promote the transformation of the apical parenchyma cells of tobacco into meristem stem cells and form adventitious buds [32]. *RcWOX1* in *Rosa canina* was found to be expressed throughout the process of callus formation, and ectopic overexpression of *RcWOX1* could significantly increase lateral root production in transgenic *Arabidopsis* [33]. In four Rosaceae plants (strawberry, peach, plum, and pear), 14, 10, 10, and 9 *WOX* genes have been identified, respectively, as well as *RoWUS* obtained in callus tissue of *Rhododendron ovatum* Planch, but no studies related to the functional validation of their genes have been seen [34,35]. *JsWOX1* and *JsWOX4* were both found to be expressed in the callus tissues of *Jasmine*, and overexpression of the *JsWOX1* gene could induce the root differentiation of the callus tissues [36]. At present, there are no reports on the WOX gene family in peony at home and abroad.

*WOX* has been researched deeply on the meristem and the development of various organs in model plants [19]. However, the molecular mechanism of the *WOX* gene family in ornamental plants, especially in the woody plant peony, is not clear. Therefore, in this study, four *PoWOX* genes were cloned from the peony genome and transcriptome data

of pre-existing peony somatic embryos. Their sequence characteristics and evolutionary relationships were analyzed, and their expression patterns were analyzed via quantitative fluorescence expression. The expression locations of these genes were determined via subcellular localization assay. Finally, the interaction proteins of the *PoWOX* genes were identified via yeast two-hybrid assay combined with bimolecular fluorescence complementation assay. The purpose of this study was to analyze the expression pattern of the *PoWOX* gene family and the molecular mechanism of the regulatory pathway in peony. It will provide a theoretical basis for obtaining an efficient in vitro regeneration and genetic transformation system of peony by molecular biology.

## 2. Materials and Methods

### 2.1. Plant Materials

The experimental material was selected from "Fengdan" (*P. ostii*), a cultivated variety formed by the long-term cultivation and evolution of the wild peony species, YangShan Peony (*P. ostii*). It has ornamental, medicinal, and oil properties and stable genetic traits. The seeds of "Fengdan" came from the peony nursery in Heze city in Shandong Province. The tissue-cultured seedlings of peony were cultured from the explants of the seed embryo. After 7 days of dark culture at $24 \pm 1$ °C, the explants were transferred to alternating culture under light for 16 h and dark for 8 h with a light intensity of 40 μmol·m$^{-2}$·s$^{-1}$ and $24 \pm 1$ °C. The successions were carried out every 7–15 days. The seeds and the root, stem, leaf, and callus tissue from the tissue-cultured seedlings of peony were quick-frozen in liquid nitrogen for fluorescence quantification experiments.

*Nicotiana benthamiana* was also grown in a lighted incubator at 24 °C with a light intensity of 40 μmol·m$^{-2}$·s$^{-1}$ and alternating light for 16 h and dark for 8 h.

### 2.2. Screening and Cloning of PoWOX Genes

Third-generation sequencing full-length transcriptome data obtained from somatic embryos "Fengdan" at various developmental stages from the preliminary work of our research group and published peony genome data were chosen as the database [37]. In addition, the *AtWOX* gene family of *A. thaliana* in the public database (TAIR10, https://www.arabidopsis.org/, accessed on 22 October 2020) was chosen as the query sequence (Table S1). The basic local alignment search tool (BLAST) was used to retrieve homologous *PoWOX* gene sequences. HMMER (hmmsearch search | HMMER (ebi.ac.uk), 27 October 2020)) was used to confirm the integrity of the HD in the candidate PoWOX proteins and remove those missing the HD domain [38]. The sequences with the correct HD were selected as the amplification sequence of the subsequent gene.

RNA was extracted from the somatic embryos of sterile peony seedlings at various developmental stages using the Quick RNA Isolation Kit (Hua Yueyang, Beijing, China). The RNA concentration was determined by spectrophotometer and the band integrity was determined by 1% agarose gel electrophoresis. The RNA was stored in a −80 °C refrigerator for later use. The cDNA was reverse transcribed using the above-extracted RNA as a template via a reverse transcription kit (A3500, Promega). Primers were designed according to the primer design principles using SnapGene software (V2.3.2) (Table 1). The cDNA was used as a template for the amplification of the target sequence via the LA Taq kit (TaKaRa, Kusatsu, Japan). In addition, the target sequence was obtained via 1% agarose gel electrophoresis when the target band matched the corresponding length of the marker sequence. The DNA products were recycled using a DNA purification and recovery kit (TIANprep Mini Plasmid Kit, DP103-03, Tiangen, Beijing, China). The target sequence was ligated to the pMD19-T vector at room temperature using the pMDTM19-T Vector Cloning Kit (TaKaRa, Kusatsu, Japan) and then transferred into DH5α *Escherichia coli*. The plasmids were extracted via a plasmid extraction kit (Tiangen, DP103), and 10 μL plasmid solution was sent for sequencing to verify the correctness of the target sequence (Anshengda, Beijing, China).

**Table 1.** Primer sequences of PCR amplification.

| Genes | Primer Sequence F (5′-3′) | Primer Sequence R (5′-3′) |
|---|---|---|
| *PoWOX1* | GCTCTAGAATGTATATGATGGGTTATAATGATGGCGGAG | TCCCCCCGGGGATTCCTCAACGGAAGGAACTCAAAATACT |
| *PoWOX4* | CGTCTAGAATGGGAAACATGAAGGTTCATCAGTTC | TCCCCCCGGGGTCTTCCTTCCGGGTGTAATGGAA |
| *PoWOX11* | GCTCTAGAATTCATGTGTTTTATCTTTTTTCTCTCAACTCT | TCCCCCCGGGGAGTGGTTCTTGAAACTAGGAAATAGCTTTC |
| *PoWOX13* | GCTCTAGAATGGGGGTTAGCAAAAAAGATTTTAGAAAGT | TCCCCCCGGGGCGAAGAATGTCAAATTCTTCCTCCTG |

### 2.3. Analysis of Physicochemical Properties and Phylogenetic Tree of PoWOX Genes

The basic physicochemical properties of the PoWOX proteins sequences were analyzed using ProtParam (ExPASy-ProtParam tool) [39]. The secondary structures of the PoWOX proteins were analyzed using SOPMA (https://npsa-prabi.ibcp.fr/cgi-bin/npsa_automat.pl?page=npsa_sopma.html, accessed on 19 May 2021)). The subcellular localizations of the PoWOX proteins were predicted using Plant-mPLoc (Plant-mPLoc server (sjtu.edu.cn, accessed on 19 May 2021)) [40].

Homology searches were performed from Ensembl Plants, the Plant Transcription Factor Database (PlantTFDB, PlantTFDB-Plant Transcription Factor Database @ CBI, PKU (gao-lab.org, accessed on 19 May 2021)) [41], and NCBI public databases (National Center for Biotechnology Information (nih.gov, accessed on 19 May 2021)). The WOX protein sequences from *A. thaliana*, *Oryza sativa*, *Juglans regia*, *Vitis vinifera*, *Populus trichocarpa*, *Amborella trichopoda*, *Theobroma cacao*, *Picea abies*, *Selaginella moellendorffii*, *Ceratopteris richardii*, *Ginkgo biloba*, *Physcomitrella patens*, and *Ostreococcus lucimarinus* were used for multiple sequence alignment using the software of Clustal W [42]. Using the MEGA 7 software with the neighbor-joining method and 1000 bootstrap replicates [43,44], the phylogenetic trees were constructed by the PoWOX protein sequences and the above WOX protein sequences of all species. Other parameters were set as default. Then, each *PoWOX* gene was given a unique name based on the complete WOX amino acid sequence according to the phylogenetic tree branching results.

### 2.4. Multiple Sequence Alignment and Conserved Structural Domain Analysis of PoWOX Genes

The multiple sequence alignment was analyzed among the amino acid sequences of WOX1, WOX4, WOX11, and WOX13 proteins from *A. thaliana*, *O. sativa*, *J. regia*, *V. vinifera*, *P. trichocarpa*, and *P. suffruticosa* using DNAMAN software. The obtained PoWOX protein sequences were submitted to Multiple Em for Motif Elicitation (MEME Version 5.4.1, http://meme-suite.org/tools/meme, accessed on 25 May 2021) for motif analysis along with the WOX protein sequences of the homologous species [45], with parameters selected to expect a motif site distribution of 0 or 1, the number of 10, and a width of 5 to 200. The MAST XML file saved and the Newick file obtained from the tree building were entered together into TBtools software under the Gene Structure View (Advanced) window for visualization and analysis [46]. The sequence identity map was drawn using the online website, WEBLOGO (http://weblogo.berkeley.edu/logo.cgi, accessed on 12 November 2021) [47].

### 2.5. Quantitative Fluorescence Analysis

Seven periods of somatic embryo development of peony (0; 5; 10; 15; 20; 30; 60 days) and five different tissues of peony (seed, root, stem, leaf, and callus) were taken and snap-frozen in liquid nitrogen. The total RNA was extracted for reverse transcription of cDNA and diluted ten times with deionized water as the template. The specific primers of quantitative fluorescence for the *PoWOX* genes were designed using Primer Premier 6.0 software (Table 2).

**Table 2.** Primer sequences of *PoWOX* genes for real-time quantitative.

| Primers Name | Primers Sequence (5′-3′) |
| --- | --- |
| Q-*PoWOX1*-F | CGTTGGCGGCAATGAAGAAGAATC |
| Q-*PoWOX1*-R | GGCAATTAGGAGGACTCAAGTTGGTAT |
| Q-*PoWOX4*-F | CCGCAACAGTCTTGGTCTTAGCC |
| Q-*PoWOX4*-R | TTCCTCATCTCTACATCTCACCTCTTCC |
| Q-*PoWOX11*-F | GCAACGCCAGATTCAAGCAAGTC |
| Q-*PoWOX11*-R | AAGAGGAACCAGCAAGACAAGAAGATG |
| Q-*PoWOX13*-F | ATGACGGACGAGCAAATAGAGGAACTT |
| Q-*PoWOX13*-R | CCGCTGCCTGGTAGTGATCTTCTG |
| Q-*Poubiquitin*-F | TCCTCCACCTCCTACCTTCCGACTC |
| Q-*Poubiquitin*-R | CGATCCTCCTGAGCCAAGCGTCAT |

Reverse transcription quantitative real-time PCR (RT-qPCR) was performed using the TB Green Premix Ex Taq II fluorescence quantification kit (Tli RNaseH Plus, TaKaRa Kusatsu, Japan) on a QTOWER real-time fluorescence PCR instrument (Analytik Jena, Germany). The PCR reaction system was as follows: TB Green Premix Ex Taq for 5 μL, cDNA template for 1 μL, primer (F+R) for 0.4 μL, and ddH$_2$O supplemented to 10 μL. Three biological replicates were set up, and the reaction conditions were 95 °C pre-denaturation for 90 s, 95 °C denaturation for 5 s, melting at 60 °C for 30 s, and 40 cycles. The melting curve was 60 to 95 °C and the temperature was increased by 1 °C every 15 s. Using peony ubiquitin ligase as the internal reference gene, the relative expressions of *PoWOX1*, *PoWOX4*, *PoWOX11*, and *PoWOX13* in each tissue were analyzed via the $2^{-\Delta\Delta CT}$ [48], with the expression of Day 0 peony somatic embryos and peony root tissue being the control group.

*2.6. Subcellular Localization Assay of PoWOX Proteins in Tobacco*

The plant expression vector was constructed by ligating the CDS sequence of *PoWOX* to the PHG vector (Shanghai Weidi Biotechnology Co., Ltd., Shanghai, China) containing the GFP tag. Restriction endonucleases (New England BioLabs, UK) BamHI and PstI were used for double digestion of the PHG plasmid to obtain single-stranded vectors. The double digestion reaction system was as follows: PHG vector for 1 μg, Cut Smart Buffer for 5 μL, BamHI for 1 μL, PstI for 1 μL, and ddH$_2$O supplemented to 50 μL. The double digestion reaction system was placed on a PCR instrument for 15 min digestion at 37 °C and 20 min heat inactivation at 65 °C. The splice primers were designed on SnapGene software (V2.3.2) (Table 3). PCR amplification was performed using the pMD19-T vector ligated to *PoWOX* as a template with the LA Taq kit's high-fidelity enzyme (TaKaRa, Kusatsu, Japan).

**Table 3.** Primer sequences of plant expression vector *35S::PoWOX-GFP*.

| Primers Name | Primers Sequence (5′-3′) |
| --- | --- |
| PHG-*PoWOX1*-F | AGTCTCTCTCTCAAGCTTGATGTATATGATGGGTTATAATGATGGC |
| PHG-*PoWOX1*-R | CGGGTCATGAGCTCCTGCAATTCCTCAACGGAAGGAACT |
| PHG-*PoWOX4*-F | AGTCTCTCTCTCAAGCTTGATGGGAAACATGAAGGTTCATCA |
| PHG-*PoWOX4*-R | CGGGTCATGAGCTCCTGCATCTTCCTTCCGGGTGTAATG |
| PHG-*PoWOX11*-F | AGTCTCTCTCTCAAGCTTGATGGAAGATCATGACCCTAACA |
| PHG-*PoWOX11*-R | CGGGTCATGAGCTCCTGCAAGTGGTTCTTGAAACTAGGAAATAG |
| PHG-*PoWOX13*-F | AGTCTCTCTCTCAAGCTTGATGGGGGTTAGCAAAAAAGATTTTAG |
| PHG-*PoWOX13*-R | CGGGTCATGAGCTCCTGCACCGAAGAATGTCAAATTCTTCCT |

The plant expression vector *35S::PoWOX-GFP* was constructed by homologous cloning of the target fragment using the 2x M5 superfast seamless cloning mix kit (Mei5 Bioservices Co., Ltd., Beijing, China). The 10 μL ligation system included 2× M5 superfast seamless cloning mix for 5 μL, the linear reaction conditions were 50 °C for 25 min. The ligated product was transformed into *E. coli* competent DH5α and incubated on LB solid medium containing kanamycin (50 μg/mL) at 37 °C for 12~16 h in inverted mode. The positive clones identified by the colony PCR were sequenced from Beijing Anshengda. Plasmids of the correctly sequenced colonies were extracted and stored at −20 °C in the refrigerator.

The plant expression vector plasmid *35S::PoWOX-GFP* was transferred into *A. tumefaciens* competent cells GV3101 (Shanghai Weidi Biotechnology Co., Ltd., Shanghai, China) by heat stimulation. The single clones were cultured for 2~3 days. The single colonies identified as positive by PCR were picked and placed into 5 mL of LB liquid medium (Kan, 50 μg/mL; Rif, 50 μg/mL; Gen, 50 μg/mL) and incubated at 28 °C with shaking at 200 rpm for 16 h. When the $OD_{600}$ of the cultured bacteria grew to 0.1, the sample was centrifuged at 4000 rpm for 10 min at room temperature to harvest the bacteria. The supernatant was discarded to collect the bacteria, which were resuspended in an infection solution (10 mM MES-KOH, 10 mM $MgCl_2$, 100 μM acetosyringone, pH 5.6) until $OD_{600} = 0.1$. The cells were washed twice and allowed to stand for 2 h at room temperature in the dark.

A 1 mL syringe (with the needle removed) was used to inject the infection solution into the back of the tobacco leaves. Then, the leaves were incubated in a 22 °C lighted incubator for 2 to 3 days. Infected tobacco leaves were cut, stained with DAPI solution for 5–10 min at room temperature, rinsed 2 to 3 times with PBS solution, and incubated for 2 to 3 days at 22 °C in a lighted incubator. The fluorescence signals of EGFP and DAPI were observed using an Axio Imager M2 microscope (Zeiss, Jena, Germany) under 509 nm and 465 nm excitation light, respectively. Then, photographs were taken.

*2.7. Yeast Two-Hybrid Assay*

2.7.1. Vector Construction

The full-length CDSs of the *PoWOX* genes were constructed into the yeast vectors pGADT7 and pGBKT7 (Clontech, USA) (Table 4). The ligated products were transformed into *E. coli* DH10B. Before plasmid extraction the positive bacteria were sequenced and compared without errors and stored at −20 °C. The amplification of the target fragment and enzymatic cleavage of the vector were performed in the same way as subcellular localization. The target fragments were cloned homologously via the Seamless Assembly Cloning Kit (Clone Smarter). A 10 μL ligating system was as follows: 2× Seamless Master Mix for 5 μL, DNA fragment and linearized vector for 2 μL, and $ddH_2O$ supplemented to 10 μL. The reaction conditions were 50 °C for 15 min. Then, the recombinant vector was transferred to Y2H Gold yeast (Hua Yueyang, Beijing, China) and incubated in an inverted incubator at 30 °C for 2~3 days.

2.7.2. Self-Activation Verification

The transformed Y2H Gold with bait pGBKT7-gene was taken and self-activation was detected on SD/-Trp and SD/-Trp-Ade-His plates incubated upside down at 30 °C for 3 days in the incubator. The bait proteins were observed in order to ensure their self-activation activity. The transformed Y2H Gold with bait pGBKT7-gene and the empty pGBKT7 were cultivated on SD/-Trp plates and incubated upside down at 30 °C for 3 days. The growth rate of the yeast was observed to determine whether the bait proteins were toxic or not.

**Table 4.** Primer sequences for the construction of pGADT7 and pGBKT7 recombinant vectors on *PoWOX* genes in Y2H.

| Primer Name | Upstream Primer F (5′-3′) | | Downstream Primer R (5′-3′) | |
|---|---|---|---|---|
| BK-PoWOX1 | ATCTCAGAGGAGGACCTGCAATGTATATGATGGGTTATAA | NdeI | AGGGGTTATGCTAGTTATGCATTCCTCAACGGAAGGAACT | NotI |
| BK-PoWOX4 | ATCTCAGAGGAGGACCTGCAATGGGAAACATGAAGGTTCA | NdeI | AGGGGTTATGCTAGTTATGCTCTTCCTTCCGGGTGTAATG | NotI |
| BK-PoWOX13 | ATCTCAGAGGAGGACCTGCAATGGGGTTAGCAAAAAAGAT | NdeI | AGGGGTTATGCTAGTTATGCCCGAAGAATGTCAAATTCTT | NotI |
| BK-PoWOX11 | ATCTCAGAGGAGGACCTGCAATGGAAGATCATGACCCTAA | NdeI | AGGGGTTATGCTAGTTATGCAGTGGTTCTTGAAACTAGGA | NotI |
| AD-PoWOX1 | GACGTACCAGATTACGCTCAATGTATATGATGGGTTATAA | NdeI | TATTAAGGGTTCCGGATCGCATTCCTCAACGGAAGGAACT | NotI |
| AD-PoWOX4 | GACGTACCAGATTACGCTCAATGGGAAACATGAAGGTTCA | NdeI | TATTAAGGGTTCCGGATCGCTCTTCCTTCCGGGTGTAATG | NotI |
| AD-PoWOX13 | GACGTACCAGATTACGCTCAATGGGGTTAGCAAAAAAGAT | NdeI | TATTAAGGGTTCCGGATCGCCCGAAGAATGTCAAATTCTT | NotI |
| AD-PoWOX11 | GACGTACCAGATTACGCTCAATGGAAGATCATGACCCTAA | NdeI | TATTAAGGGTTCCGGATCGCAGTGGTTCTTGAAACTAGGA | NotI |

### 2.7.3. Yeast Co-Transformation

pGBKT7-gene and pGADT7-gene were co-transformed into Y2H Gold strain to study the protein interactions between PoWOX transcription factors. The constructed pGBKT7-gene and pGADT7-gene vector plasmids were separately co-transformed into the Y2H Gold strain at 500 ng each as the experimental group. pGBKT7-53 and pGADT7-T were co-transformed as the positive control group, and pGBKT7-LAM and pGADT7-T were co-transformed as the negative control group. A total of 10 µL of each bacterial solution was diluted 10-fold with 0.9% NaCl solution and then applied on SD/-Trp-Leu plates and incubated upside down in an incubator at 30 °C for 3 days. Monoclonal clones were picked and dissolved in sterile ddH$_2$O and diluted 10-, 100-, and 1000-fold. A total of 10 µL of every diluted solution was dropped on SD/-Trp-Leu and SD/-Trp-Leu-Ade-His/X-α-gal/Aureobasidin (SD/-Trp-Leu-Ade-His/X-α-gal/AbA) plates and incubated in an inverted incubator at 30 °C for 3 days. The growth conditions were observed and photographed with a Stemi 305 microscope (Zeiss, Jena, Germany).

### 2.8. Bimolecular Fluorescence Complementation Experiments

According to the bimolecular fluorescence complementation (BiFC), the restriction sites were selected according to the vector sequences of pSPYNE(R)173 (YNE) and pSPYCE(MR) (YCE). The primers were designed using SnapGene software (V2.3.2) (Table 5). The target fragments were amplified using the method of subcellular localization, and the methods of enzymatic cleavage and ligation of the linear vector were used for yeast two-hybrid assay (Y2H).

The procedure for tobacco leaf infection in the BiFC is referred to the subcellular localization. In this experiment, it was necessary to calculate the volume of the required bacterial solution so that the final OD$_{600}$ was 0.5. The bacterial was resuspended with bacteria solution and the bacterial solution was combined two by two according to the experimental group in the required amount. The bacterial solution was injected with a 1 mL syringe into the abaxial surface of 5- to 6-week-old robust and rich green, thick tobacco leaves, which were then incubated for 2 to 3 days in a lighted incubator. The infested tobacco leaf parts were cut and stained with DAPI solution for 5–10 min and rinsed with PBS solution 2 to 3 times. Then the EGFP and DAPI fluorescence signals were observed and photographed using a Zeiss Axio Imager M2 microscope under 509 nm and 465 nm excitation light, respectively.

**Table 5.** Primer sequences for the construction of pSPYNE(R)173 and pSPYCE(MR) recombinant vectors on *PoWOX* genes in BiFC.

| Gene Name | Upstream Primer F (5′-3′) | | Downstream Primer R (5′-3′) | |
|---|---|---|---|---|
| YCE-PoWOX1 | TTACGCTGGGCCCAGGCCTAATGTA TATGATGGGTTATAA | SpeI | CGGTACCCTCGAGGTCGACGATTC CTCAACGGAAGGAACT | BamHI |
| YCE-PoWOX4 | TTACGCTGGGCCCAGGCCTAATGGG AAACATGAAGGTTCA | SpeI | CGGTACCCTCGAGGTCGACGTCTT CCTTCCGGGTGTAATG | BamHI |
| YCE-PoWOX13 | TTACGCTGGGCCCAGGCCTAATGGG GTTAGCAAAAAAGAT | SpeI | CGGTACCCTCGAGGTCGACGCCG AAGAATGTCAAATTCTT | BamHI |
| YCE-PoWOX11 | TTACGCTGGGCCCAGGCCTAATGGA AGATCATGACCCTAA | SpeI | CGGTACCCTCGAGGTCGACGAGT GGTTCTTGAAACTAGGA | BamHI |
| YNE-PoWOX1 | GGATCTTGGGCCCAGGCCTAATGTA TATGATGGGTTATAA | SpeI | CGGTACCCTCGAGGTCGACGATTC CTCAACGGAAGGAACT | BamHI |
| YNE-PoWOX4 | GGATCTTGGGCCCAGGCCTAATGGG AAACATGAAGGTTCA | SpeI | CGGTACCCTCGAGGTCGACGTCTT CCTTCCGGGTGTAATG | BamHI |
| YNE-PoWOX13 | GGATCTTGGGCCCAGGCCTAATGGG GTTAGCAAAAAAGAT | SpeI | CGGTACCCTCGAGGTCGACGCCG AAGAATGTCAAATTCTT | BamHI |
| YNE-PoWOX11 | GGATCTTGGGCCCAGGCCTAATGG AAGATCATGACCCTAA | SpeI | CGGTACCCTCGAGGTCGACGAGT GGTTCTTGAAACTAGGA | BamHI |

## 3. Results and Analysis

### 3.1. Cloning and Phylogenetic Analysis of PoWOX Genes in Peony

Using *AtWOX* gene family sequences of *A. thaliana* as query sequences, a total of four *PoWOX* gene sequences were screened from the peony genome database and the third-generation full-length transcriptome database of peony was obtained by the research group earlier [37]. Four transcripts were identified with complete HD by HMMER and belonged to the WOX gene family. The CDS sequences of the four *PoWOX* genes, named *PoWOX1*, *PoWOX4*, *PoWOX11*, and *PoWOX13*, were amplified and obtained using the total cDNA from different developmental periods of peony somatic embryos.

The basic physicochemical properties of the four obtained *PoWOX* genes were analyzed, including protein length, relative molecular weight, isoelectric point, instability coefficient, average hydrophilicity number, and aliphatic index (Table 6). The results revealed that the length range of the PoWOX proteins was 211~317 aa and the relative molecular weight was about 24~37 kDa. All the proteins were unstable hydrophilic proteins. Subcellular localization prediction of the PoWOX proteins found that all the PoWOX proteins were localized in the nucleus (Table 6). The secondary structure of the PoWOX proteins consisted of α-helix, β-sheet, extended chain, and random coil, and the structures of these amino acids were "helix-loop-helix-turned-helix" (Table 7).

**Table 6.** Basic physicochemical properties of *PoWOX* genes.

| Gene | GenBank Accession | ORF (bp) | Protein Length (aa) | MW (Da) | pI | Instability Coefficient | GRAVY | Aliphatic Index | Subcellular Localization |
|---|---|---|---|---|---|---|---|---|---|
| *PoWOX1* | OM457047 | 951 | 317 | 36562.68 | 8.71 | 55.93 | −0.93 | 60.63 | Nucleus |
| *PoWOX4* | OM457050 | 633 | 211 | 24300.55 | 9.36 | 47.58 | −0.91 | 60.52 | Nucleus |
| *PoWOX11* | OM457049 | 765 | 255 | 28165.35 | 6.13 | 67.96 | −0.46 | 61.88 | Nucleus |
| *PoWOX13* | OM457048 | 858 | 286 | 32543.31 | 4.92 | 64.70 | −0.80 | 69.27 | Nucleus |

Notes: ORF: open reading frame; bp: Base pair; aa: amino acid; MW: molecular weight; Da: Dalton; pI: isoelectric point; GRAVY: Grand average of hydropathicity.

To further understand the evolutionary relationships of the WOX gene family of peony, the four obtained *PoWOX* genes were compared with a total of 123 sequences from 13 species by multiple sequences and an evolutionary tree was constructed (Table S1). The

results revealed that all WOX members were divided into three clades: modern/WUS clade (WC-WOX), intermediate clade (IC-WOX), and ancient clade (AC-WOX). Among them, PoWOX1 and PoWOX4 were in the modern clade, which were more closely related to *T. cacao* and *P. trichocarpa*, respectively. PoWOX11 was in the intermediate clade, which was more closely related to *V. vinifera*. PoWOX13 was in the ancient clade, which was more closely related to *A. trichopoda*. WC-WOX only contains gymnosperms and angiosperms generally, but we detected that CrWOXB and CrWUS of *C. richardii* are also distributed in the modern branch, which may represent an evolutionary stage of transition from IC-WOX (Figure 1).

**Table 7.** Predicted secondary structure of PoWOX proteins.

| Protein | α-Helix | β-Sheet | Extended Chain | Random Coil |
|---------|---------|---------|----------------|-------------|
| PoWOX1 | 29.65% | 12.62% | 3.15% | 54.57% |
| PoWOX4 | 25.41% | 6.63% | 4.42% | 63.54% |
| PoWOX11 | 20.11% | 15.76% | 9.24% | 54.89% |
| PoWOX13 | 39.51% | 8.39% | 3.15% | 48.95% |

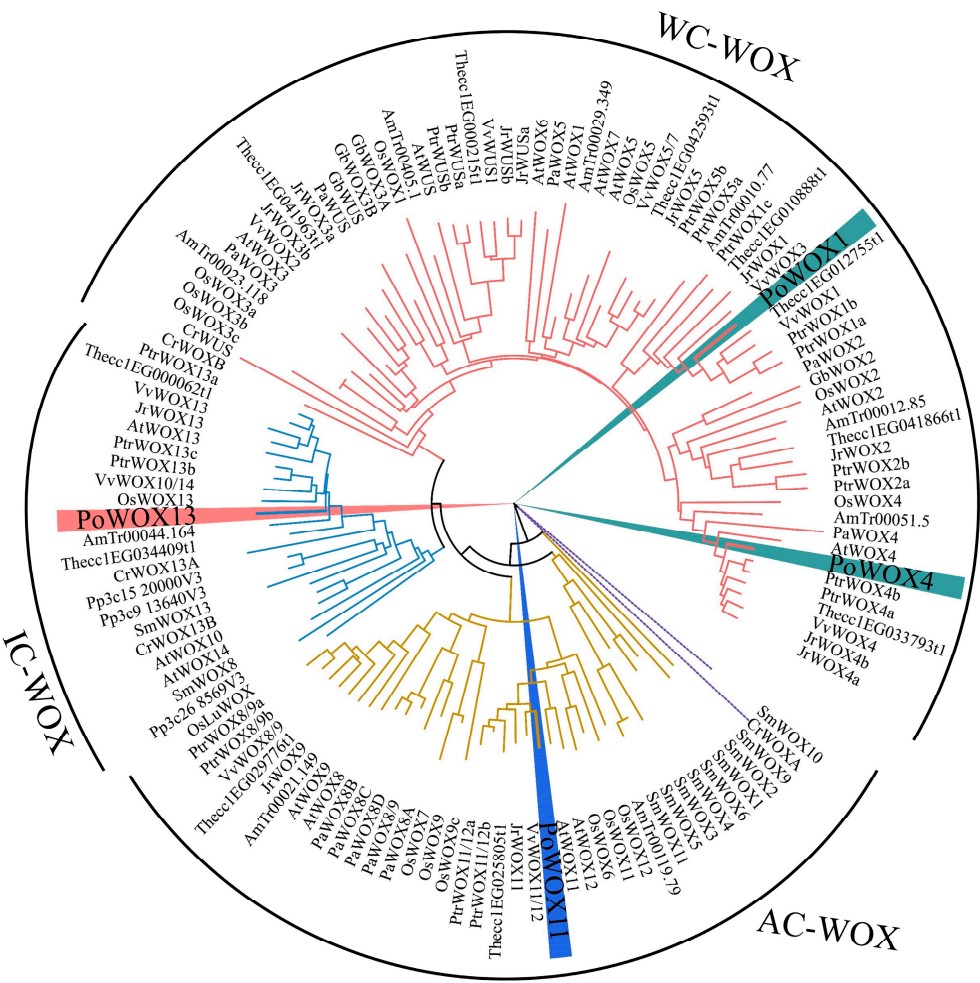

**Figure 1.** Phylogenetic analysis of *WOX* genes in plants. Evolutionary analysis of the full-length sequences of WOX proteins was performed by the NJ method using MEGA 7. A total of 127 WOX sequences from 14 species were classified into three clades: modern clade (WC-WOX), intermediate clade (IC-WOX), and ancient clade (AC-WOX). PoWOX1 and PoWOX4 were highlighted with green long triangular tags; PoWOX11 was highlighted with dark blue long triangular tags; PoWOX13 was highlighted with red long triangular tags.

### 3.2. Multiple Sequence Alignment and Sequence Characterization Analysis of PoWOX Proteins in Peony

The multiple sequence alignment was analyzed among the four PoWOX proteins and the WOX1, WOX4, WOX11, and WOX13 proteins distributed the same subclade from the model species *A. thaliana*, *O. sativa*, and three closely related species, namely, *J. regia*, *V. vinifera*, and *P. trichocarpa*. The results showed that PoWOX proteins also contain HD and they were highly conserved (Table S1). The results of the homologous structural domain comparison showed that the highly conserved residue sites in the HD included R, W, P, Q, L, I, G, V, F, and N (Figure 2).

A phylogenetic tree was built for over 31 WOX sequences (Figure 3a), and it was revealed that they were still divided into three clades, WC-WOX, IC-WOX, and AC-WOX, where PoWOX1 and PoWOX4 clustered in the WC-WOX, PoWOX11 in the IC-WOX, and PoWOX13 in the AC-WOX. Based on the amino acid sequences, the conserved motifs of the six species (*A. thaliana*, *O. sativa*, *P. suffruticosa*, *J. regia*, *V. vinifera*, and *P. trichocarpa*) were analyzed and a total of 10 conserved motifs were identified (Figure 3b). Among them, blue motif 1 was HD, which was presented in all branches. Red motif 5 was the WUS motif with sequence TLELFPLH, which was presented only in the WC-WOX including WOX1 and WOX4 (except JrWOX1). For most of the remaining motifs, no function has been characterized yet. Yellow motif 2 was highly conserved and was presented in every clade. Dark gray motif 6 and light green motif 7 were unique to WOX4 proteins and may exercise unique functions in WOX4 proteins. Purple motif 9 was presented in WOX4 sub-branches and AtWOX13. Orange motif 10 and light gray motif 3 were only present in IC-WOX, suggesting that they may have special functions in vascular plants. Dark green motif 4 and orange motif 8 also presented separately in AC-WOX.

### 3.3. Analysis of the Expression Pattern of PoWOX Genes in Peony

To verify the function of the *WOX* gene in peony seed embryo formation and tissue development, the expression pattern of *PoWOX* genes in peony was analyzed by RT-qPCR. According to the expression pattern of *PoWOX* in the peony seed embryo at different stages of development in vitro (Figure 4), the expression level of *PoWOX1* was highest at 0 days of somatic embryo development and dropped to the lowest at 30 days. Then, the expression level increased abruptly at 60 days. It indicated that *PoWOX1* plays a role in the whole period of development of the peony somatic embryo. The expression level of *PoWOX4* was highest at 15 days of somatic embryo development. The results of RT-qPCR also showed that *PoWOX11* and *PoWOX13* presented similar expression patterns, with the highest expression in leaves at 5 days of somatic embryo development and the lowest expression at 60 days. According to the expression pattern of *PoWOX* in different tissues of peony seedlings (Figure 5), it was found that *PoWOX1* had the highest expression level in leaves, followed by seeds and callus. *PoWOX4* in callus obtained the highest level, whereas *PoWOX11* in seeds obtained the highest. The expression level of *PoWOX13* in seeds was the highest and in roots it was higher.

### 3.4. Subcellular Localization of PoWOX Proteins

Prediction of the PoWOX proteins by the Plant-mPLoc server revealed that the PoWOX1, PoWOX4, PoWOX11, and PoWOX13 proteins were localized in the nucleus (Table 6). In this study, four *35::PoWOX-GFP* fusion expression vectors were constructed using PHG plant expression vectors to detect the subcellular localization of the PoWOX proteins. It was found that the positive control *35S::GFP* fusion proteins had fluorescent signals in both the nucleus and cytoplasm, and the green fluorescent signals of all the PoWOX proteins were observed only in the nucleus of epidermal cells in tobacco (Figure 6). This provided further validation that all four PoWOX proteins are localized in the nucleus, and it is speculated that the PoWOX proteins may function as transcription factors. Meanwhile, subcellular localization experiments showed that the PoWOX proteins have common expression sites in cells, which could be detected further for protein interactions.

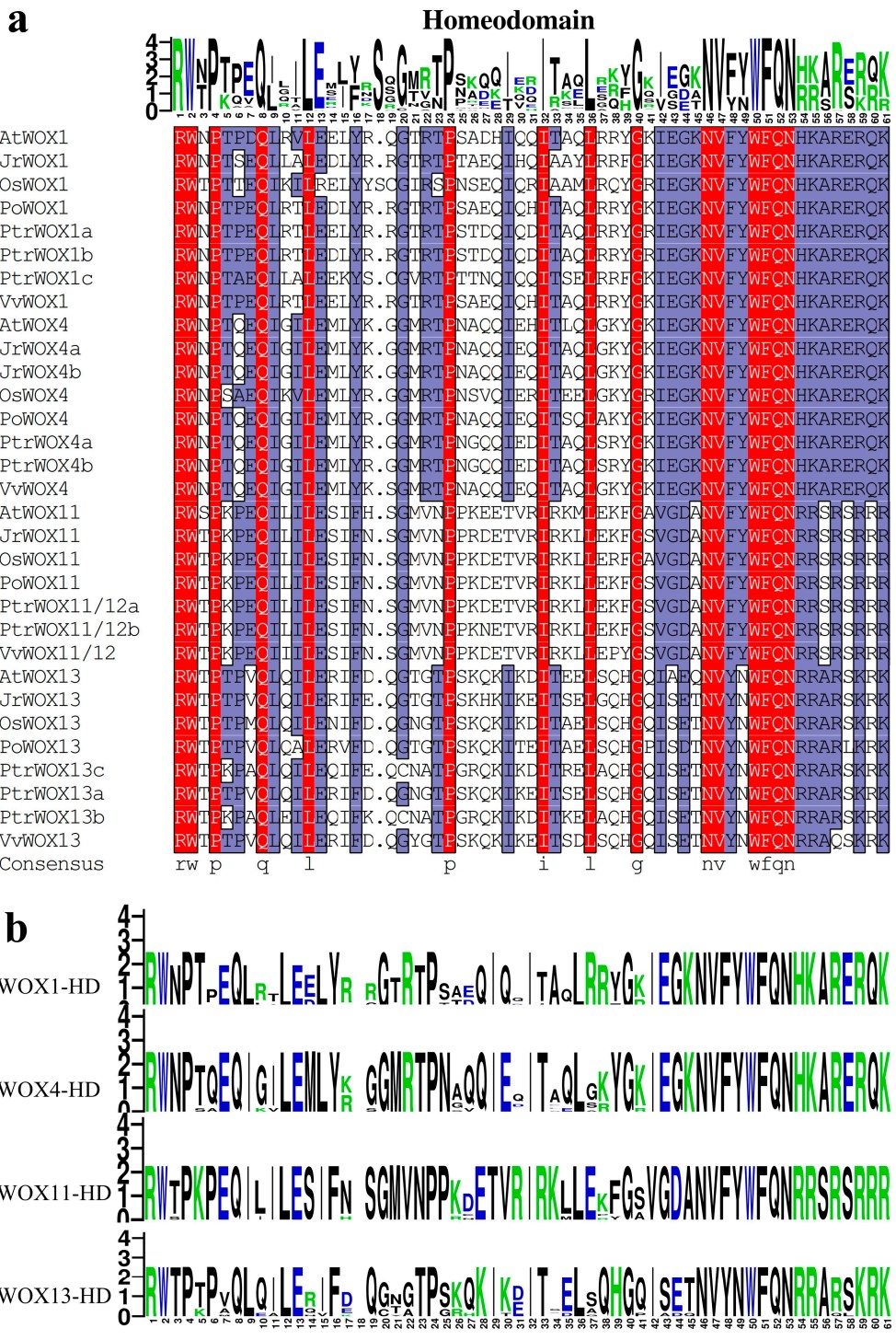

**Figure 2.** Multiple sequence alignment and sequence identification of WOX1, WOX4, WOX11, and WOX13 proteins. (**a**) Multiple sequence alignment of WOX1, WOX4, WOX11, and WOX13 proteins in six species (*A. thaliana*, *O. sativa*, *J. regia*, *V. vinifera*, *P. trichocarpa* and *P. suffruticosa*). The amino acid sequence similarity indicated in red was 100%, and the amino acid sequence similarity in purple was greater than or equal to 50%. (**b**) Sequence identification map of WOX-HD.

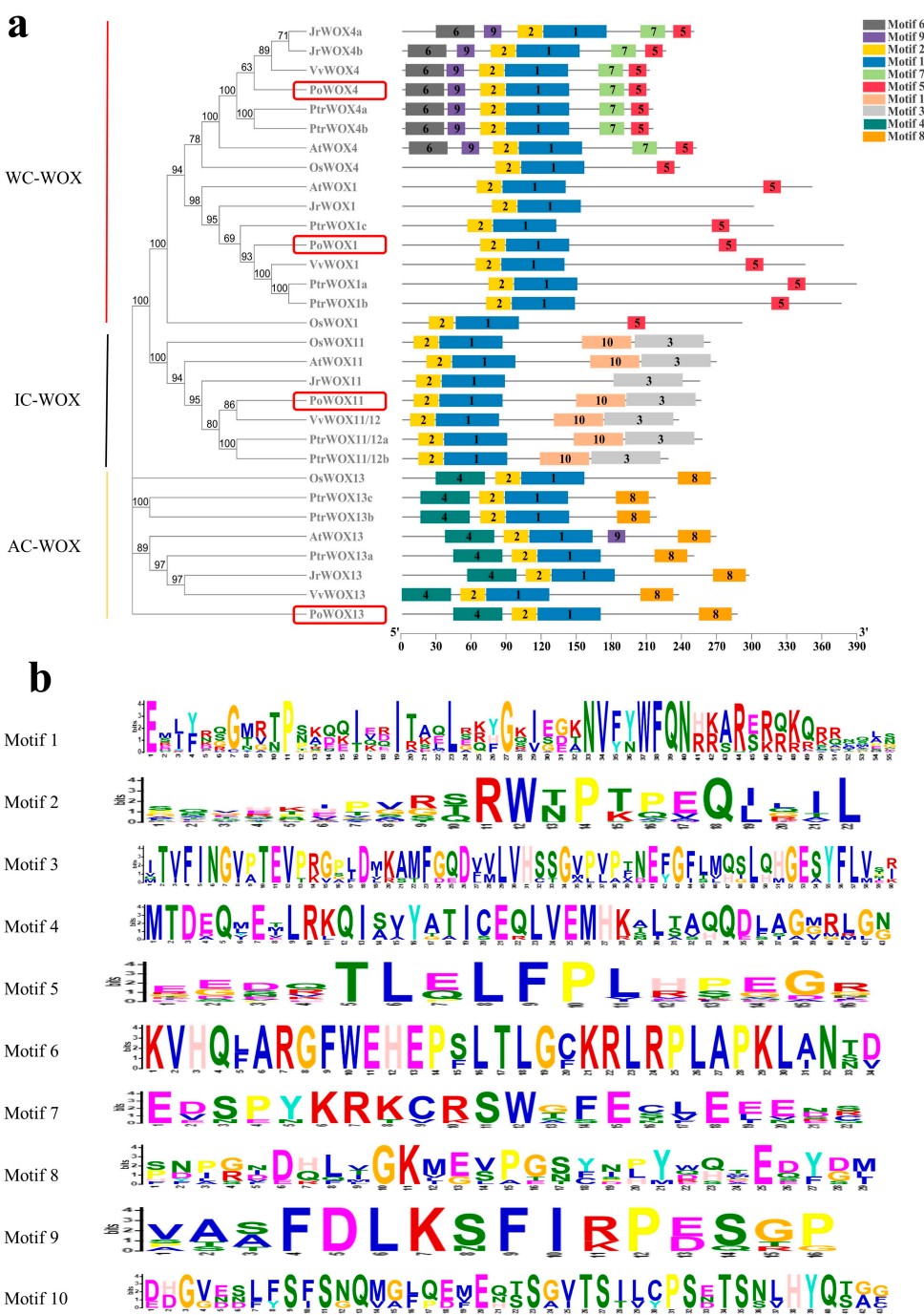

**Figure 3.** Conserved motif analysis of WOX proteins and sequence identification of ten conserved motifs. (**a**) the phylogenetic tree and conserved motif analysis of six species (*A. thaliana*, *O. sativa*, *P. suffruticosa*, *J. regia*, *V. vinifera*, and *P. trichocarpa*), the red rectangle circles the peony PoWOX protein. A total of ten conserved motifs were identified, and each motif was represented by a separate number and color. Blue motif 1 represents HD, red motif 5 represents that the WUS motif is only present in the WC-WOX, yellow motif 2 is present in every clade, dark gray motif 6 and light green motif 7 are unique to WOX4 protein, purple motif 9 is present in the WOX4 subclade and AtWOX13, orange motif 10 and light grey motif 3 belong to the IC-WOX, and dark green motif 4 and orange motif 8 exist alone in the AC-WOX. (**b**) Sequence identification of the predicted 10 motifs of WOX protein.

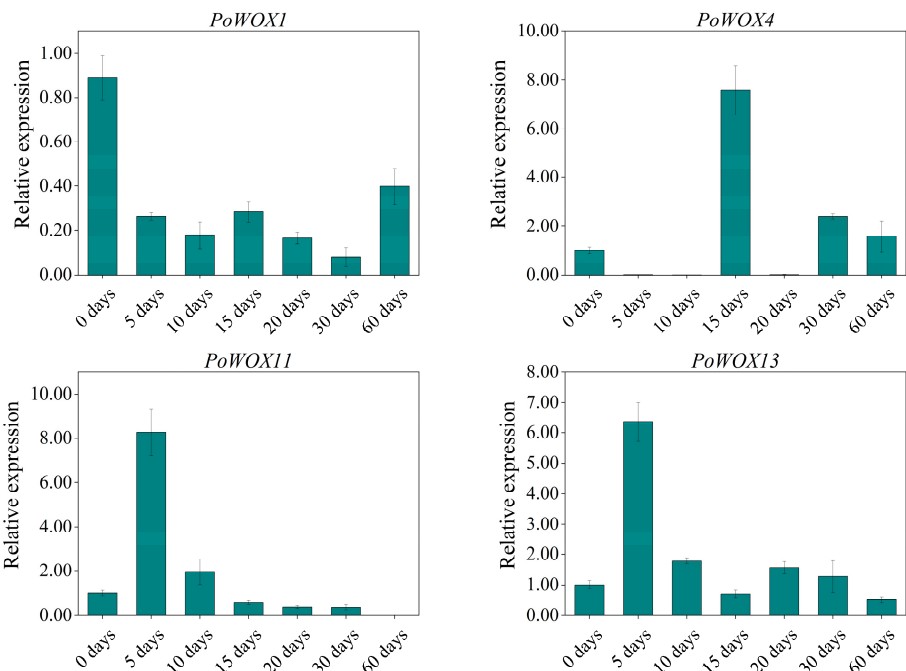

**Figure 4.** Expression of *PoWOX* genes of peony in the developmental stage of somatic embryos. Expression levels of peony somatic embryos in seven developmental stages (0; 5; 10; 15; 20; 30; 60 days). The expression of Day 0 peony somatic embryos was the control group.

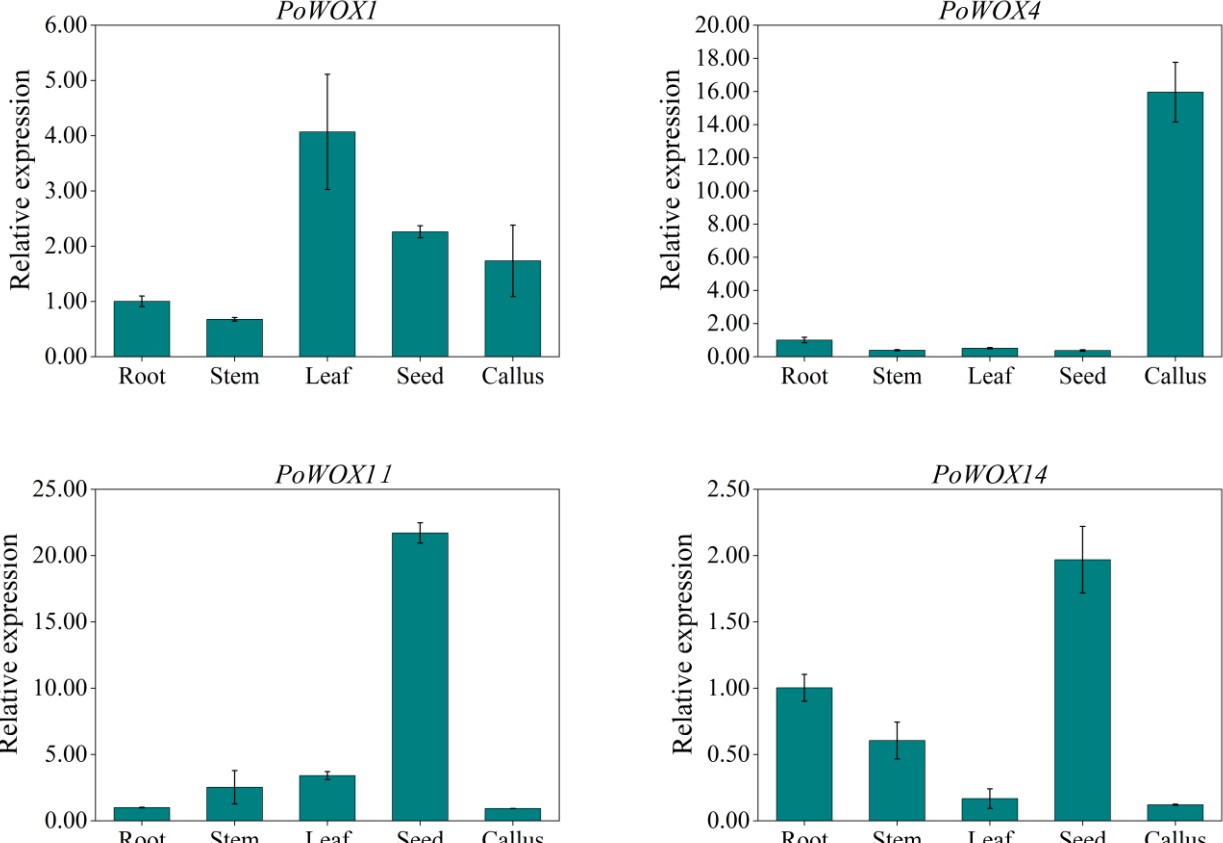

**Figure 5.** Expression of *PoWOX* genes in peony tissues. The expression of *PoWOX* gene in five different of peony (seed, root, stem, leaf, and callus). The expression of peony root tissues was the control group, respectively.

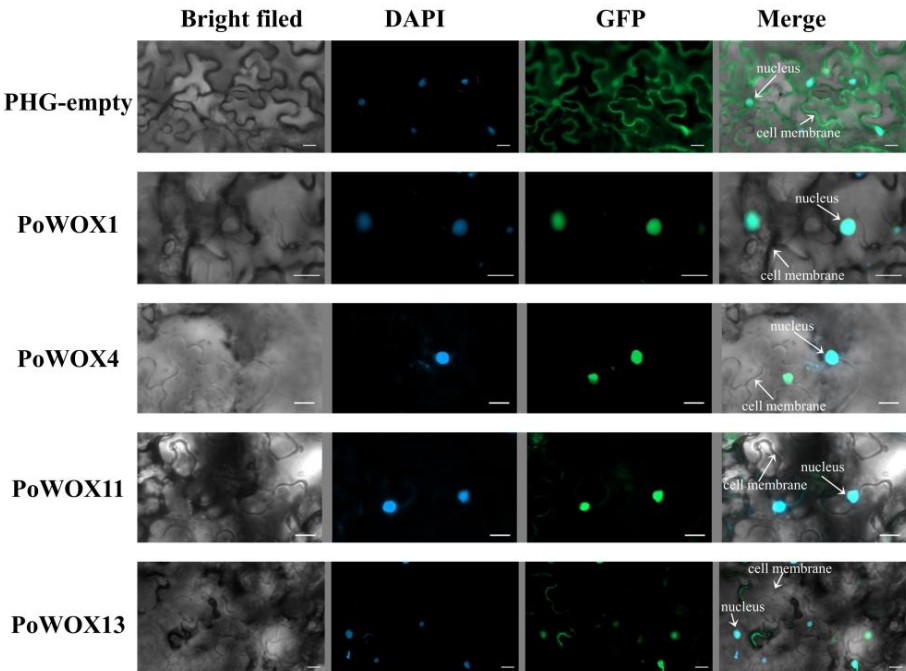

**Figure 6.** Subcellular localization analysis of *35::PoWOX-GFP* fusion proteins. *The 35::PoWOX-GFP* vector was infected with *N. benthamiana* mediated by *Agrobacterium* GV3101, and the expression of the fusion protein was observed under the Zeiss Axio Imager M2 microscope. Positive control *35S::GFP* fusion protein was expressed in the nucleus and cytoplasm, and PoWOX protein was expressed in the nucleus. Tobacco leaf cells were stained. Green fluorescence shows the fusion protein location and blue fluorescence is the nuclear signal localized by DAPI for bright field, DAPI, GFP, and merge, respectively. Positive control is *35S::GFP* empty. The white arrows indicate the location of the nucleus and the cell membrane. The scale bar is 20 μm.

### 3.5. Results of Interactions between PoWOX Proteins

The Y2H-Gold-GAL4 yeast two-hybrid system was used to identify protein interactions between PoWOX1, PoWOX4, PoWOX11, and PoWOX13. First, the PoWOX proteins were tested for toxicity and self-activation. Their CDS sequences were cloned into the pGBKT7 vector as baits and then transferred into Y2H Gold yeast cells and cultured on SD/-Trp plates. It was noticed that the yeast strains containing the pGBKT7-WOX vector could all grow normally on SD/-Trp plates and did not differ from the control group in terms of growth. This indicates that the recombinant vectors were not toxic and did not affect the normal growth of the yeast strain. In addition, the pGBKT7-WOX1 and pGBKT7-WOX13 vector strains could not grow on the SD/-Trp-Ade-His plates, but pGBKT7-WOX4 and pGBKT7-WOX11 could (Figure 7). This showed that the PoWOX1 and PoWOX13 proteins do not have the ability to self-activate and, therefore, could be used as both bait and prey for the subsequent yeast interaction experiments. However, PoWOX4 and PoWOX11 proteins can only be used as prey proteins for co-transformation experiments because of their self-activating ability.

According to the above results, eight combinations of the yeast two-hybrid assays were performed using PoWOX1 and PoWOX13 as bait and prey, and PoWOX4 and PoWOX11 as prey in two-by-two interactions. All normally growing yeasts on non-selective SD/-Trp-Leu plates showed that all bait vector pGBKT7-WOX and prey vector pGADT7-WOX plasmids were successfully transferred into the Y2H Gold strain (Figure 8). Then, based on the analysis of the growth condition of the co-transformed yeast strains on selective SD/-Trp-Leu-Ade-His/X-α-gal/AbA plates, PoWOX11 could interact as prey to form heterodimers with PoWOX1 and PoWOX13, and PoWOX1 and PoWOX13 could also form homodimers. As a prey protein, PoWOX4 could not form dimers with other PoWOX proteins, and the rest of the combinations did not have direct interactions.

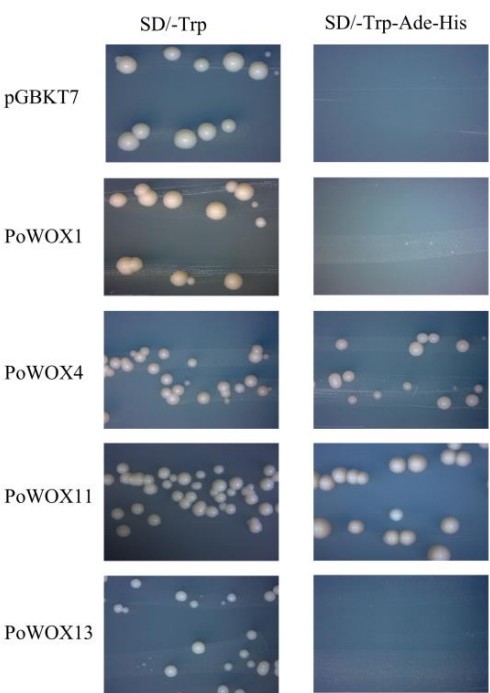

**Figure 7.** Yeast self-activating analysis. All yeasts grew normally on SD/-Trp plates, and colonies of PoWOX4 and PoWOX11 appeared on SD/-Trp-Ade-His plate.

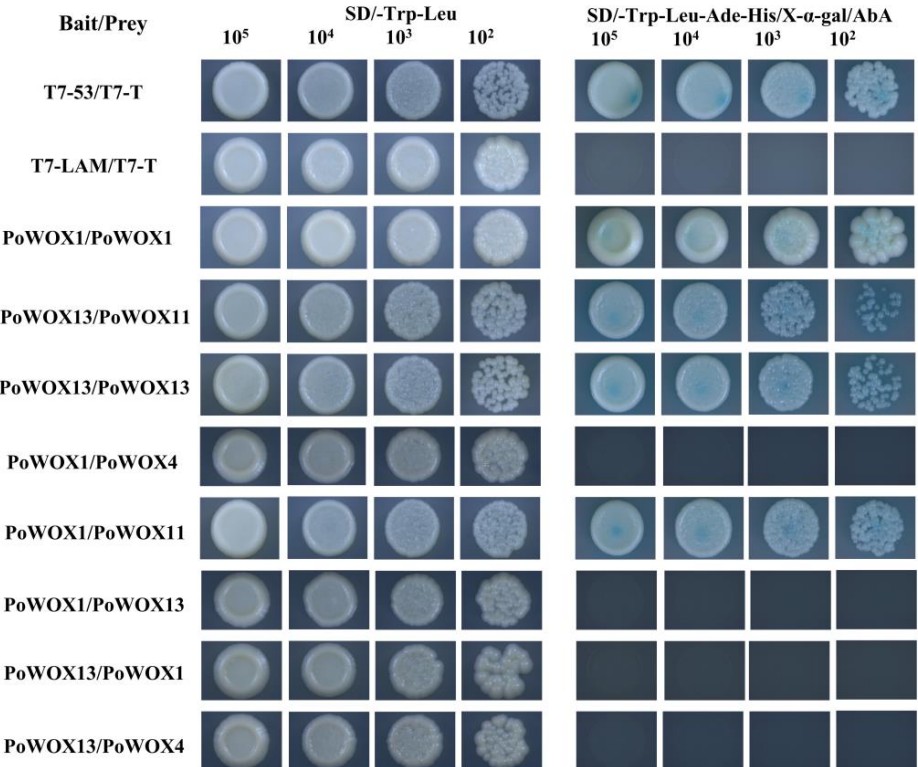

**Figure 8.** Protein interactions analysis between PoWOX members in yeast two-hybrid assay. The CDS sequences of PoWOX1, PoWOX4, PoWOX11, and PoWOX13 were cloned into pGBKT7 (bait) vector and pGADT7 (prey) vector, respectively. Co-transformation to SD/-Trp-Leu plates and SD/-Trp-Leu-Ade-His/X-α-gal/AbA plates was performed to verify the interactions between PoWOX members. pGBKT7-53/pGADT7-T (T7-53/T7-T) was co-transformed as a positive control and pGBKT7-LAM/pGADT7-T (T7-LAM/T7-T) was co-transformed as a negative control. The pictures were taken by Zeiss Stemi 305 microscope to preserve.

### 3.6. BiFC to Verify the Interactions between PoWOX Proteins

The interactions between the PoWOX proteins were verified by BiFC experiments. The fluorescence signals were visualized under the microscope based on the results of previous Y2H co-transformation experiments in two-by-two combinations. The results showed that green fluorescent signals were observed for the experimental groups YCE-PoWOX1/YNE-PoWOX1, YCE-PoWOX1/YNE-PoWOX11, YCE-PoWOX11/YNE-PoWOX13, and YCE-PoWOX13/YNE-PoWOX13 (Figure 9), while the negative control group had no green fluorescent signal. This inferred that there is an interaction relationship between the four groups of PoWOX proteins, PoWOX1–PoWOX1, PoWOX1–PoWOX11, PoWOX11–PoWOX13, and PoWOX13–PoWOX13.

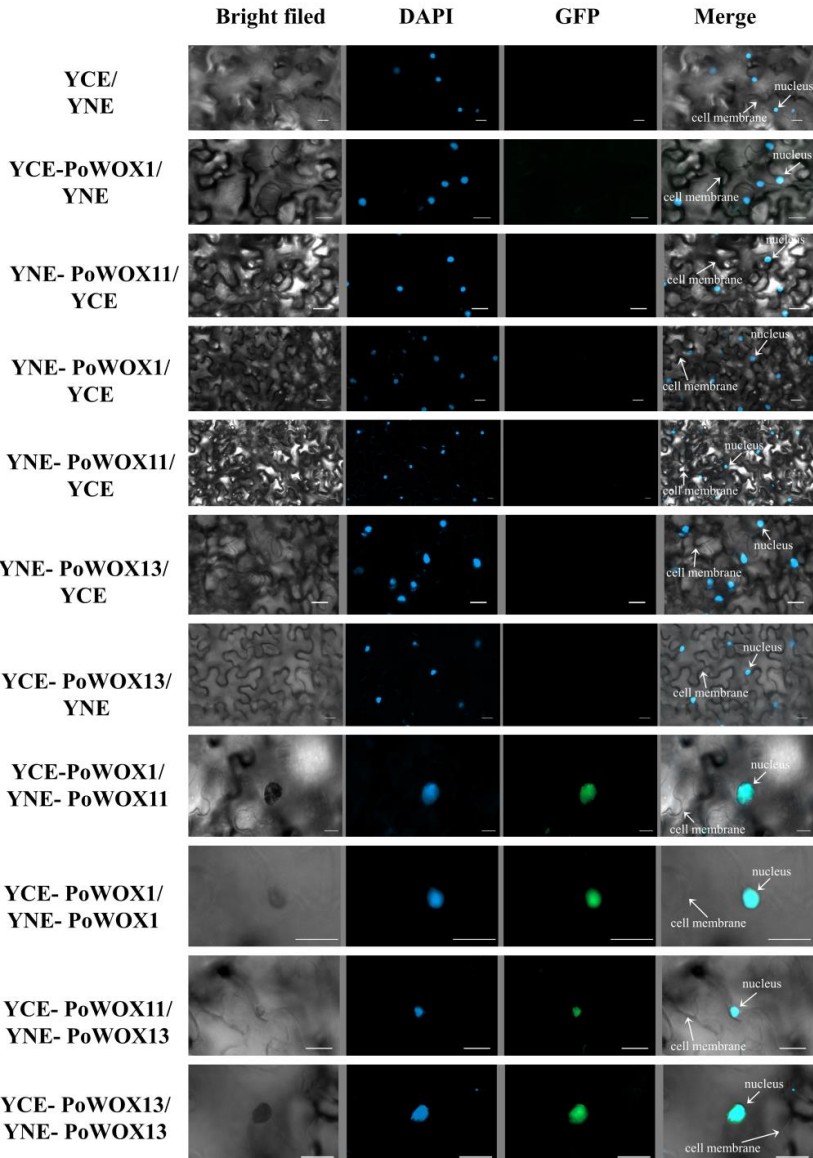

**Figure 9.** Subcellular localization analysis of *35::PoWOX-GFP* fusion protein. *35::PoWOX-GFP* vector was used to infiltrate the lower epidermis of tobacco mediated by *A. tumefaciens* GV3101. Fusion protein expression was observed under Zeiss Axio Imager M2 microscope. Tobacco leaf cells were stained with DAPI (1 μg/mL). Green fluorescence shows the fusion protein location and blue fluorescence is the nuclear signal localized by DAPI for bright field, DAPI, GFP, and merge, respectively. Positive control is *35S::GFP* empty. The white arrows indicate the location of the nucleus and the cell membrane. The scale bar is 20 μm.

## 4. Discussion

### 4.1. Evolution Analysis of PoWOX Gene Family

The *WOX* gene family is associated with the diversity of stem cell types during plant evolution. From low mosses and lycopodiopbyta plants to gymnosperms and angiosperms, stem cells play an important role in individual development. The conserved function of the *WOX* gene family in plant evolution corroborates that the origin of the stem cell regulation mechanism is ancient and conservative in evolution [49]. For example, the presence of WOX homologs in green algae, the oldest species in the plant kingdom, suggests that the WOX family may have originated in green algae. In the moss *P. patens*, *PpWOX13L* (*WOX13* homologous gene) was found to play an important regulatory role in the initiation and maintenance of stem cells [50]. The expression of *WOX* genes was also shown to be involved in the regeneration process of organs in lycophytes [51].

From the phylogenetic tree (Figure 1), it can be seen that the ancient clade of the plant WOX family contains at least one WOX member from the lower plants (green algae and mosses) to the vascular plants (lycophytes and euphyllophytes). However, the WOX members of the non-vascular plants green algae and mosses are only presented in the ancient clade. The members of the vascular plants only contained in the intermediate clade of the WOX family imply the specificity on the IC-WOX. The study suggested that the IC-WOX is specific to vascular plants and that extant vascular plants are descendants of two early ancestral lineages, the euphyllophytes lineage, which included ferns and seed plants, and the lycophytes lineage [52]. From the evolutionary tree it can be seen that IC-WOX is present in lycophytes (*S. moellendorffii*), extant ferns (*C. richardii*), gymnosperms (*G. biloba* and *P. abies*), and angiosperms (*A. trichopoda*, *A. thaliana*, *J. regia*, *V. vinifera*, *T. cacao*, *P. trichocarpa*, *O. sativa*, and *P. suffruticosa*). It has been suggested that the evolution of *WOX* genes may accompany the adaptation of plants from aquatic to terrestrial environments. In general, *WOX* is a relatively old gene family involved in plant evolution. However, current studies on the gene functions of WOX family members in seed plants, especially in woody plant stem cells, are still unclear. The four *PoWOX* genes cloned from peony were distributed among the ancient, intermediate, and modern clades of the WOX family, providing an entry point for subsequent studies on stem cell diversification in peony.

### 4.2. Expression Pattern Analysis of PoWOX Gene Family

The fluorescence quantification of the *PoWOX* genes showed different expression patterns in different tissues of peony. According to the expression patterns of *PoWOX* at the various stages of somatic embryo development in vitro (Figure 4), it was observed that *PoWOX1* had the highest expression at 0 days and *PoWOX4* at 15 days. Both *PoWOX11* and *PoWOX13* were the most highly expressed at 5 days. It was hypothesized that *PoWOX* may be expressed in the stem apical meristem, which could improve the differentiation and development of the vegetative organs of peony by enhancing the activity of shoot apical stem cells. Meanwhile, according to the expression pattern of *PoWOX* genes in different tissues of peony (Figure 5), *PoWOX1* obtained the highest expression level in leaves, followed by seeds and callus. This showed that the *PoWOX1* gene may be related to leaf morphogenesis. *PoWOX4* with its highest expression level in callus may promote callus proliferation or differentiation. *PoWOX11* and *PoWOX13* both had the richest expression in seeds and were considered to play a major function in seed development. At the same time, the expression of *PoWOX13* in roots was second only to that in seeds, and it is speculated that *PoWOX13* may be involved in both seed and root development.

The functions of *WOX* genes may be variable in annuals and woody plants [12]. In *Arabidopsis*, *AtWOX4* mainly promotes the development of the primitive cambium and modulates vascular cell division [24,26]. *PoWOX4*, however, was highly expressed in the callus, which may facilitate the proliferation or differentiation of peony callus. Gene duplication and differentiation may result in an increase in the number of *WOX* gene families in different species. Therefore, there are generally functional redundancies in *WOX* gene families and the different expression patterns of *WOX* genes in various species.

The WOX protein family participates in the maintenance and differentiation of all stem cells in plants, including the primary meristem (shoot apical meristem (SAM) and root apical meristem (RAM)) and the secondary meristem (vascular meristem), through similar or specific regulatory networks. It has been demonstrated that WOX family members can express from postembryonic development to lateral organs in plants and have functions in embryonic development, vascular formation, and leaf morphogenesis [17,19,53]. Further elucidation of the regulatory mechanism of the WOX gene family in peony is beneficial to solve the bottlenecks of tissue culture, such as low induction rate of callus and difficult rooting in peony, in molecular breeding.

*4.3. PoWOX Genes May Form a Dimer to Play a Transcriptional Regulatory Role in Peony Growth and Development*

Different from the embryonic development of animals, all new organs of many plants develop during postembryonic development. In addition, the new organs are derived from the division and differentiation of the apical meristem and various external environmental factors and internal genetic influences [54]. Plant stem cells located in the central zone of the apical meristematic tissue have the capacity of self-renewal and multiple differentiation and are the core of the development regulation processes of plants. The organization center provides the important microenvironmental signals for stem cell development and is necessary for maintaining stem cell function [55]. It has been shown that CLV3, the dodecapeptide secreted from stem cell expression, can receive the signals from CLV1, which is a receptor kinase rich in the leucine. In addition, CLV3 with the members of the CLV1 family can downregulate the expression of WUS from the *WOX* gene family together. The WUS protein can be expressed in the organization center and move to the apical part of the meristematic tissue to activate the stem cell marker gene *CLV3* and maintain the properties of stem cells. The expression of WUS and CLV3 together constitutes a negative feedback regulatory loop [56–58].

The WOX transcription factor family is a subfamily of the homeobox (HOX) transcription factor superfamily, and it encodes the homeodomain with 60–66 amino acids [13]. The typical secondary structure of WOX proteins consists of α-helix, β-fold, extended chain, and random coil. These amino acids form a structure, named "helix-loop-helix-turn-helix", to help the interactions between mediated proteins. Therefore, WOX transcription factors are prone to forming homodimers or heterodimers. These dimers were found to promote the regulation of stem cell activity from the WUS protein movement to stem cells through plasmodesmata [57]. Studies of the mutation, deletion, and substitution of WUS protein sequences have revealed that the homeodomain of WUS and the non-structural region between it and the WUS-box play an important role in WUS homodimerization [57,59]. The interactions between WUS and DNA could be stabilized by the WUS dimers to combine with the different DNA motifs in order to activate and repress the expression of the downstream genes of *WUS* directly or indirectly [60]. In this work, we demonstrated that the PoWOX1 and PoWOX13 proteins in the peony WOX family can form homodimers by yeast two-hybrid and BiFC assays individually. Peony stem cell activity may be regulated from PoWOX1 and PoWOX13 by forming dimers and moving to peony stem cells through plasmodesmata like the WUS proteins in *Arabidopsis*.

As transcription factors, WUS proteins can play a regulatory role by interacting with other proteins. The heterodimer can be formed by the interaction between WUS proteins in *Arabidopsis* and STM. It co-binds to the promoter of *CLV3* in order to enhance the binding strength between WUS and the *CLV3* promoter, activating the expression of the *CLV3* gene, and enhancing stem cell activity at the stem apex [61]. WUS proteins and the family members of transcription factors HAIRY MERISTEM (HAM) can interact to promote stem cell proliferation and maintain the homeostasis of stem cells in the SAM [62]. It has been found that WOX4 and WOX5 of the WOX family can interact with HAM proteins, and the regulation of WUS-HAM-CLV3 on the polarity of plant apical meristem tissue was demonstrated by a combination of computer models and experimen-

tal evidence [63]. By using the yeast two-hybrid assay and the BiFC assay, it has been shown that the formation of rice epidermal hairs can be modulated by the interaction between *OsWOX3B* and the AP2/ERF transcription factor *Hairy Leaf 6* (*HL6)* in rice [64]. Subsequently, a protein complex was formed to enhance the binding between *HL6* and the growth hormone-related gene *OsYUCCA5* [64]. In this research, PoWOX11 interacted with PoWOX1 and PoWOX13 to form heterodimers. The two pairs of interacted proteins, PoWOX11–PoWOX1 and PoWOX11–PoWOX13, may play important regulatory functions in promoting the proliferation of stem cells and maintaining the homeostasis of stem cells in the SAM of peony stems.

At present, an efficient in vitro regeneration and genetic transformation system of peony has not been reported, hindering the process of peony genetic engineering breeding. By exploring the critical genes and regulatory factors in the development of the peony somatic embryo, the bottleneck of peony genetic transformation would be broken by improving the efficiency of in vitro regeneration of peony using molecular means. This is beneficial not only to understand the molecular and regulatory mechanisms of peony somatic embryo development but also to achieve directed breeding and improvements in efficiency through genetic engineering breeding technology to accelerate the fundamental process of molecular breeding in peony.

## 5. Conclusions

In this research, four *PoWOX* genes, named *PoWOX1*, *PoWOX4*, *PoWOX11*, and *PoWOX13*, were cloned from the somatic embryo in peony "Fengdan" by PCR and identified by bioinformatics. *PoWOX1*, *PoWOX4*, *PoWOX11*, and *PoWOX13* were shown to have vital functions in the development of peony somatic embryos, and the expression of each gene in different parts of peony was tissue specific. The *PoWOX* were all localized in the nucleus by subcellular localization assays. Using yeast two-hybrid and BiFC methods, it was shown that the PoWOX1 and PoWOX13 proteins could form homodimers by themselves, and PoWOX11 interacted with PoWOX1 and PoWOX13 to form heterodimers. Different protein complexes may play roles in different regulatory pathways, such as peony stem cell maintenance and meristem development.

**Supplementary Materials:** The following supporting information can be downloaded at: https://www.mdpi.com/article/10.3390/horticulturae8030266/s1. Table S1: The accessions and sequences of WOX proteins used in Figure 1. Table S2: The WOX-HD multiple sequence comparison results sequence used in Figure 2.

**Author Contributions:** Conceptualization, M.X., W.Z., Z.J. and T.H.; methodology, M.X., Y.M. and Y.D.; software, M.X. and Y.C.; validation, Y.C. and Y.D.; data curation, K.F. and X.Z.; writing—original draft preparation, M.X. and W.Z.; writing—review and editing, M.X., W.Z. and T.H. All authors have read and agreed to the published version of the manuscript.

**Funding:** This research was supported by ICBR Fundamental Research Funds Grant (NO. 1632019009), ICBR Fundamental Research Funds Grant (NO. 1632021005) and ICBR Fundamental Research Funds Grant (NO. 1632020001).

**Institutional Review Board Statement:** Not applicable.

**Informed Consent Statement:** Not applicable.

**Data Availability Statement:** All data in this study could be found in the manuscript or Supplementary Materials.

**Acknowledgments:** We thank all the colleagues that helped with the development of different parts of this manuscript.

**Conflicts of Interest:** The authors declare no conflict of interest.

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
