# Peer review of "A Preliminary Investigation on the Functional Validation and Interactions of PoWOX Genes in Peony (Paeonia ostii)"

_horticulturae, doi:10.3390/horticulturae8030266_

Round 1

Reviewer 1 Report

Xia et al have developed this manuscript by identifying and functionally validating PoWOX genes in Peony. However, writing errors has made it difficult to review the manuscript. It is recommended that authors improve the writing errors before it can be peer-reviewed.

Abstract:

  1. Confusing sentence, restructure and rephrase it: It will be an urgent problem in peony molecular breeding to establish an efficient and stable in vitro regeneration and genetic transformation system in order to overcome the recalcitrant characteristics of peony regeneration and shorten the breeding cycle.
  2. Why is it necessary to mention: Plant-specific WUSCHEL-related homeobox

It is not easy to review without the line numbers. Include line number in each page to provide comments for reference

Figure 4: Remove d from x-axes and include the label as days

Figure 6 and 9: Use arrows to label the cellular structures

There are numerous writing errors throughout the manuscript that makes it difficult to review. It is recommended that authors correct all language and grammatical errors before the manuscript can be peer reviewed

Author Response

Response to Reviewer 1 Comments

Thank you for your valuable and thoughtful comments. And we have carefully checked and improved the English writing in the revised manuscript. The detailed corrections were listed below point by point:

Point 1:

Abstract: Confusing sentence, restructure and rephrase it:

It will be an urgent problem in peony molecular breeding to establish an efficient and stable in vitro regeneration and genetic transformation system in order to overcome the recalcitrant characteristics of peony regeneration and shorten the breeding cycle.

Response 1:

Thank you for the reminder. Our original writing will indeed cause ambiguity, now according to your opinion, the corresponding sentences in the abstract have been modified as follows :

There is an urgent need in peony molecular breeding to establish an efficient and stable in vitro regeneration and genetic transformation system, in order to overcome the recalcitrant characteristics of peony regeneration and shorten the breeding cycle.

Point 2:

Why is it necessary to mention: Plant-specific WUSCHEL-related homeobox.

Response 2:

From the references, we can know that the WUSCHEL-related homeobox (WOX) genes form a plant-specific subclade of the eukaryotic homeobox transcription factor superfamily, which is characterized by the presence of a conserved DNA-binding homeodomain, and this subclade is not present in animals. Therefore, we refer to WOX as a plant-specific transcription factor. The corresponding references are Ref. 12 and Ref. 13, as follows:

  1. van der Graaff, E.; Laux, T.; Rensing, S. A. The WUS Homeobox-Containing (WOX) Protein Family. Genome Biol. 2009, 10 (12), 248, doi:10.1186/gb-2009-10-12-248.
  2. Haecker, A.; Gross-Hardt, R.; Geiges, B.; Sarkar, A.; Breuninger, H.; Herrmann, M.; Laux, T. Expression Dynamics of WOX Genes Mark Cell Fate Decisions during Early Embryonic Patterning in Arabidopsis Thaliana. Development 2004, 131 (3), 657–668, doi:10.1242/dev.00963.

Point 3:

It is not easy to review without the line numbers. Include line number in each page to provide comments for reference.

Response 3:

We are very sorry about this, as the missing line numbers do make the review more difficult and inconvenient. Therefore, following your suggestion, we have added consecutive line numbers to each page of the manuscript. This is to make it easier for reviewers to quickly and easily point out problems when commenting on our manuscript.

Point 4:

Figure 4: Remove d from x-axes and include the label as days.

Response 4:

We have modified the label on the X-axis by removing the "d" from the X-axis and changing the "d" to "days" in the associated text description.

Revised Figure 4 and its notes (lines 414~421):

Figure 4. Expression of PoWOX genes of peony in the developmental stage of somatic embryos. Expression levels of peony somatic embryos in seven developmental stages (0; 5; 10; 15; 20; 30; 60 days). The expression of Day 0 peony somatic embryos was the control group. There were 3 biological replicates, using peony ubiquitin ligase as the internal reference gene.

Point 5:

Figure 6 and 9: Use arrows to label the cellular structures.

Response 5:

Based on your recommendation, in order to make the meaning of the pictures more visual and accurate, we have added white arrows at the appropriate places on the pictures to indicate the location of the nucleus and cell membrane, and have given the corresponding explanations in the figure notes. Please see the details in Figure 6 (lines 440~450) and Figure 9 (lines 498~507).

Point 6:

There are numerous writing errors throughout the manuscript that makes it difficult to review. It is recommended that authors correct all language and grammatical errors before the manuscript can be peer reviewed.

Response 6:

We have improved the English writing in the revised manuscript through MDPI's language services based on the reviewers' comments. The manuscript has been carefully revised and the English language has been re-scrutinized through the language polishing service, including checking the grammar, spelling, punctuation, and phrasing of the manuscript. Please see the blue revision marks in the manuscript.

Please see the attachment for the revised manuscript.

Reviewer 2 Report

Dear Author

The manuscript entitled “A preliminary investigation on the functional validation and interactions of PoWOX genes in Peony (Paeonia ostii)” was revised. The manuscript can be published after a few correction. 

Kind regards

Reviewer 3 Report

The manuscript treats about an important issue in in vitro regeneration of peony plants involving regulatory factors such as WOX transcription factors. Resolving the expression of WOX genes during  the development of peony embryos will a crucial step in the regulation of embryo development in woody plants such as peony. Because of sufficient molecular methods used in the manuscript and sufficient interpretation of obtained results, I recommend the paper to publication.

Author Response

Response to Reviewer 3 Comments

Thanks very much for the carefully review and kind comments. We wish you all the best!

Round 2

Reviewer 1 Report

In figure 4, "d" should have been removed the labels in the figure and add days in x-axes, but authors removed from the caption. Please revise.

I do not have any other comments.
